# The Role of Insulin Resistance in Fueling NAFLD Pathogenesis: From Molecular Mechanisms to Clinical Implications

**DOI:** 10.3390/jcm11133649

**Published:** 2022-06-24

**Authors:** Rossella Palma, Annamaria Pronio, Mario Romeo, Flavia Scognamiglio, Lorenzo Ventriglia, Vittorio Maria Ormando, Antonietta Lamazza, Stefano Pontone, Alessandro Federico, Marcello Dallio

**Affiliations:** 1Department of General Surgery and Surgical Specialties ‘Paride Stefanini’, Sapienza University of Rome, 00161 Rome, Italy; annamaria.pronio@uniroma1.it; 2Hepatogastroenterology Division, Department of Precision Medicine, University of Campania “Luigi Vanvitelli”, Piazza Luigi Miraglia 2, 80138 Naples, Italy; mario.romeo@unicampania.it (M.R.); flavia.scognamiglio@unicampania.it (F.S.); lorenzo.ventriglia@unicampania.it (L.V.); alessandro.federico@unicampania.it (A.F.); marcello.dallio@unicampania.it (M.D.); 3Gastroenterology and Endoscopy Unit, AORN San Giuseppe Moscati, 83100 Avellino, Italy; ormandov@yahoo.it; 4Department of Surgery Pietro Valdoni, Sapienza University of Rome, 00161 Rome, Italy; antonietta.lamazza@uniroma1.it; 5Department of Surgical Sciences, Sapienza University of Rome, 00161 Rome, Italy; stefano.pontone@uniroma1.it

**Keywords:** non-alcoholic fatty liver disease, insulin resistance, precision medicine

## Abstract

Non-alcoholic fatty liver disease (NAFLD) represents a predominant hepatopathy that is rapidly becoming the most common cause of hepatocellular carcinoma worldwide. The close association with metabolic syndrome’s extrahepatic components has suggested the nature of the systemic metabolic-related disorder based on the interplay between genetic, nutritional, and environmental factors, creating a complex network of yet-unclarified pathogenetic mechanisms in which the role of insulin resistance (IR) could be crucial. This review detailed the clinical and pathogenetic evidence involved in the NAFLD–IR relationship, presenting both the classic and more innovative models. In particular, we focused on the reciprocal effects of IR, oxidative stress, and systemic inflammation on insulin-sensitivity disruption in critical regions such as the hepatic and the adipose tissue, while considering the impact of genetics/epigenetics on the regulation of IR mechanisms as well as nutrients on specific insulin-related gene expression (nutrigenetics and nutrigenomics). In addition, we discussed the emerging capability of the gut microbiota to interfere with physiological signaling of the hormonal pathways responsible for maintaining metabolic homeostasis and by inducing an abnormal activation of the immune system. The translation of these novel findings into clinical practice could promote the expansion of accurate diagnostic/prognostic stratification tools and tailored pharmacological approaches.

## 1. Introduction

Non-alcoholic fatty liver disease (NAFLD) represents the major cause of liver disease worldwide and is classically defined by an excessive hepatic fat accumulation identified using imaging or histology in the absence of secondary causes of liver steatosis such as significant alcohol consumption, long-term use of steatogenic medications, or heredity [1]. NAFLD includes two pathologically distinct conditions with different prognoses, non-alcoholic fatty liver (NAFL) and non-alcoholic steatohepatitis (NASH), and encompasses a wide spectrum of disease severity, including fibrosis, cirrhosis, and hepatocellular carcinoma (HCC) [2].

The prevalence of NAFLD is increasing at approximately the same rate as obesity [3]. Currently, the global prevalence of NAFLD in the general population has been estimated at 25% whereas the global prevalence of NASH ranges from 3% to 5% [4]. In this context, the burden of NAFLD-related HCC is increasing dramatically. Patients with NAFLD-associated HCC had a 1.2-fold higher risk of death within 1 year, as compared to patients with HCCs of other aetiologies—especially older patients with lower incomes and unstaged tumors [5]. In a holistic view of the disease, a group of experts reached a consensus that NAFLD did not reflect current knowledge, and metabolic (dysfunction)-associated fatty liver disease (MAFLD) was suggested as a more appropriate term [6]. In this line, it has been demonstrated that the presence of metabolic syndrome (MS), especially obesity and insulin resistance (IR), can increase the rate of liver fibrosis progression, leading to cirrhosis, HCC, and/or death. NAFLD pathogenetic mechanisms are still unclarified, but a high-calorie diet, excess (saturated) fats, refined carbohydrates, high fructose intake, and a “Western” diet have all been associated with weight gain and obesity and, consequently, NAFLD.

A key role in the pathogenetic mechanisms is played by IR through reductions in whole-body, hepatic, and adipose tissue insulin sensitivity; IR may enhance hepatic fat accumulation by increasing free fatty acid delivery and by the effect of hyperinsulinemia to stimulate anabolic processes [7]. Despite its metabolic nature, many extrahepatic manifestations have been associated with NAFLD in clinical practice, such as obstructive sleep apnea, chronic kidney disease, and osteoporosis [8].

In this paper, we reviewed the supporting mechanisms of IR, both the traditional and more innovative, as a common denominator, focusing on potential premature diagnosis and therapy targets. Particularly, in addition to the classic influences, we aimed to describe the emerging findings supporting the genetic, epigenetic, and hormonal factors impacting IR status along with the novel implications of the gut microbiota and systemic immune response in the genesis and progression of IR-related hepatic steatosis in NAFLD. On the basis of this, we indicated the current status concerning the potential management frontiers for this disease.

## 2. Molecular Links between Insulin Resistance and NAFLD Pathogenesis: What Is New?

### 2.1. Insulin Resistance, Oxidative Stress and Inflammation: Molecular Mechanisms Linking the Three Vertices of the Same Triangle

IR is classically defined as a reduced systemic biological response to insulin signaling through receptor pathway activation, impairing glucose uptake and associated with other metabolic consequences particularly relevant in recognized insulin-sensitive areas such as the adipose tissue and liver [9]. Glycolipid homeostasis is widely sustained by the tangled biological network between the adipose and hepatic tissues, whose dysfunction configures a dysmetabolic context able to trigger NAFLD onset and promote its progression [10].

Furthermore, in the adipose tissue, insulin physiologically exerts an anabolic role, causing the repression of lipolysis and the increase in lipogenesis, especially during the post-prandial time frame. Conversely, in insulin-resistant individuals, the alteration of the regulatory pathways fuels lipolysis, culminating in the release of free fatty acids (FFAs) that, consequently, reach the liver and cause a fat overload in the hepatocytes [11].

FFAs’ hyper-afflux induces mitochondrial dysfunction, causing the incomplete oxidation of fatty acids (FAO) and worsening the hepatic-IR; in addition, it enhances gluconeogenesis, repressing the insulin-dependent glycogen synthesis [12]. In the liver, the lack of suppression of hepatic glucose production (HGP) represents, together with the induction of the de novo lipogenesis (DNL), the pathogenetic cornerstone for NAFLD development, and both can be attributed, at least in part, to the impairment of the molecular pathway downstream of the insulin receptor (I-Rec) [13].

Interestingly, the I-Rec pathway activation is responsible for the regulation of the hepatic lipid metabolism through the sterol-regulatory element-binding protein-1c (SREPB1c) [14], a transcription factor whose main role is represented by the upregulation of genes involved in fatty acid biosyntheses, such as acetyl-CoA carboxylase (ACC), fatty acid synthase (FAS), adenosine triphosphate (ATP) citrate lyase, and fatty-acid-elongase complex [15].

When NAFLD-related hepatic IR occurs, the inhibition of the above-mentioned I-Rec molecular pathway results, as expected, in the lack of hepatic gluconeogenesis inhibition, remaining unexpectedly the DNL, unaltered or even increased (Figure 1) [16]. To overcome this paradox, several findings were initially proposed by the concept of pathway-selective hepatic insulin resistance, supporting a dichotomic dysregulation of insulin signaling through the inhibition of the Akt/FoxO1 and the maintenance of the SREBP1c biological activity [17,18].

However, Vatner et al., in a mouse model of high-fat diet (HFD) induced NAFLD, subsequently disavowed this hypothesis, proposing the existence of different synergic alternative mechanisms responsible for the concomitant increase in DNL and gluconeogenesis in insulin-resistant liver tissue, including the hyperinsulinemia, insulin-independent re-esterification of adipose tissue-derived FFAs as well as the enhanced generation of acetyl CoA via pyruvate carboxylase (PC) activation [19,20]. A recent study of Horst et al., comparing a group of NAFLD-obese patients with only obesity affected patients, revealed an increase in the hepatic flow of lipogenic substrates, elevated 24-h plasma glucose concentrations, and a considerable insulin concentration as factors directly related to the increase in hepatic DNL without pathway-selective hepatic IR [21].

In addition, the molecular analysis of NAFLD-obese patients’ liver biopsies revealed the constitutive activation of carbohydrate response element-binding protein (ChREBP) as a consequence of the increase in intrahepatic carbohydrates and hepatic exposure to lipogenic substrates [21].

Activated ChREBP induced the expression of several genes encoding necessary components of the glycolytic pathway, resulting in a further increase in metabolic precursors for DNL [22]. In addition, the upregulation of ChREBP induced the expression of stearoyl-CoA desaturase 1 (Scd1), an enzyme involved in the biosynthesis of monounsaturated fatty acids (MUFAs), representing a further mechanism responsible for the increase in liver fat content [23].

The above-presented evidence designed the classic pathogenetic model in which IR unilaterally fuels NAFLD. Currently, as recently reported by Bugianesi et al., it is still not clear whether steatosis represents the cause or the consequence NAFLD-associated IR [24]. In a modern view, in the optic also of the identification of potential pharmacological targets, NAFLD and IR could be considered as the “two faces of Janus” [24]. For this purpose, a growing number of emerging findings, according to the theory of NAFLD as a key driver of MS [25], have suggested the bilateral correspondence between NAFLD and IR: the steatosis may influence the occurrence and worsen hepatic and systemic IR, thus creating a vicious cycle that contributes to the maintenance and progression of the disease, though the mechanisms for this are not completely understood [24,26,27].

In this scenario, IR has a close and mutual relationship with oxidative stress and inflammation that is the foundation for the worsening clinical picture and immune dysfunction [28]. In particular, the mitochondrial dysfunction classically occurring in liver steatosis could be pivotally responsible, among other consequences, for the incomplete oxidation of fatty acids (FAO), the production of reactive oxygen species (ROS), and the generation of intermediate toxic lipids (lipoperoxidation products), that, in turn, enhance the ROS generation, promote local inflammation, and alter insulin signaling [29]. In the context of the steatosis, phlogosis is largely maintained by the overexpression of the nuclear factor-kB transcription factor (NF-κB) through the upregulation of the inhibitor of nuclear factor-kappa β (IKKβ). The translocation of NF-κβ in the nucleus induces the production of inflammatory mediators such as tumor necrosis factor-α (TNF-α), interleukin-6 (IL-6), and interleukin-1β (IL-1β), favoring the recruitment and activation of Kupffer cells, which in turn, through the activation of the suppression of the cytokine-signaling (SOCS) pathway, aggravate the degree of IR by inhibiting IRS1 and IRS2 (Figure 1) [30,31].

In addition, the recruitment of hepatic macrophages determines the secretion of profibrogenic mediators, such as the transforming growth factor-β (TGF-β), representing a critical event for the fibrogenic response in the progression from NAFLD to NASH and from NASH to fibrosis and cirrhosis.

Consistently, in this context, IR appears as a “deus ex machina", able to promote fibrosis onset in different ways. Dongiovanni et al. showed IR’s ability to promote the extracellular matrix (ECM) deposition through a mechanism triggered by lipotoxicity that involved the overexpression of lysyl oxidase-like 2 (LOXL2) [32]. Relevantly, the hepatic upregulation of LOXL2 was only found in NAFLD patients with diabetes mellitus type 2 and in fibrotic progression, making this enzyme a promising therapeutic target [33]. In addition, a limited number of papers have explored the direct activation of hepatic stellate cells (HSCs) under hyperinsulinemic and hyperglycemic conditions. One of the pioneering studies conducted by Svegliati-Baroni et al. highlighted that IR-related hyperinsulinemia could directly induce the proliferation of HSCs and the secretion of type I collagen via the aberrant activation of PI3K- and ERK-dependent pathways [34]. Recently, Villar-Lorenzo et al. showed that the interaction between insulin-like growth factor 1 (IGF1) and its IGF1-receptor in HSCs triggered the phosphorylation of ERK1/2, which led to the activation of the transcription factor AP-1. Once translocated to the nucleus, AP-1 positively regulated the expression of the matrix metalloproteinase (MMP) 9, responsible for fibrotic deposition [35]. Overall, IR favors the genesis and progression of NAFLD, which in turn appears to worsen hepatic and systemic IR status. Oxidative stress and inflammation play a key and bidirectional role in an intricate network of pathogenetic mechanisms.

To further complicate this classic scenario, a plethora of modern findings, which will be analyzed in the next section of this review, supports emerging elements (such as gut dysbiosis and innate immune dysfunction) as additional tiles in the mosaic of pathogenetic IR-related NAFLD.

### 2.2. Gut Microbiota and Kupffer Cells Influence on Insulin Resistance in NAFLD: A Novel Pathogenetic Frontier

Over the last decade, the gut microbiota has acquired a predominant role in the pathogenesis of several metabolic diseases [36].

In NAFLD, the alteration of the composition combined with the reduction in the intestinal microbial species richness, defining the status of dysbiosis, has been frequently reported [37]. Consistently, the evidence has also revealed a strong association between NAFLD and small intestinal bacterial overgrowth (SIBO) [38,39,40]. Recently, Mikolasevic et al., by assessing SIBO through the esophagogastroduodenoscopy aspiration of the descending duodenum, demonstrated a higher incidence of advanced NAFLD spectra [41]. Furthermore, a previous work pointed out significantly higher serum endotoxin levels in NAFLD patients with SIBO, suggesting a key pathogenetic role [42]. The establishment of intestinal dysbiosis characterized a critical event responsible for the impairment of gut-barrier function, becoming a catalyst for metabolic endotoxemia and disruption of multiple metabolic molecular pathways [43]. Over time, studies have focused on the close relationship between the intestine and the liver, introducing the concept of the gut–liver axis involved in the pathogenesis of several disturbances [44,45,46]. Through the portal circulation, intestinal blood and its contents reach the liver, resulting in the induction or progression of liver damage [47]. Clinically, this may be reflected in the occurrence of a systemic low-grade inflammation status, potentially responsible for dysmetabolic IR-related-NAFLD-associated conditions including obesity, hyperglycemia, and dyslipidemia [48]. In this sense, a milestone finding was discovered in a large study with a cohort of 292 individuals, including non-obese and obese patients, that revealed significant differences in the microbiota composition in the obese group [49].

A data analysis evidenced the association of low bacterial richness (low gene count (LGC)) with NAFLD, IR, and low-grade inflammation as well as the domination of pro-inflammatory bacteria (e.g., *Ruminococcus gnavus* or *Bacteroides*), rather than anti-inflammatory bacteria (e.g., *Faecalibacterium prausnitzii*) [49].

Several studies showed that the advent of dysbiosis induced an alteration of the proteins of the intestinal tight junctions, such as zonula occludens-1 and occluding [50,51,52]. Moreover, Miele et al. reported a decrease in the expression of the zonula occludens-1 protein in 35 NAFLD patients, which could be responsible for the increased intestinal permeability as compared to healthy individuals [53].

The establishment of the high mucosal permeability determines the release into the systemic circulation of pathogen-associated molecular patterns (PAMPs), including bacterial DNA and lipopolysaccharide (LPS), a component of the Gram-negative outer membrane, as well as products derived from bacterial fermentation of fibers as short-chain fatty acids (SCFAs), including acetate, propionate, and butyrate [54]. LPS and other PAMPs were involved in the activation of the hepatic innate immune system via the Toll-like receptor (TLR) signaling cascade [55]. TLRs are a highly conserved family of pattern-recognition receptors (PRRs) expressed on the surface of Kupffer cells, hepatocytes, biliary epithelial cells, endothelial cells, and dendritic cells [56]. LPS reaches the liver through the portal circulation by binding to a specific protein, namely LPS-binding protein (LBP). The LPS/LBP complex, after specifically acknowledging cells expressing the cluster of differentiation 14 (CD14), interacts with the TLR4 [57]. This interaction determines the approach of the adapter molecule myeloid differentiation factor 88 (MyD88), which is responsible for the activation of several signaling pathways, such as mitogen-activated protein kinase (MAPK) and NF-kB [58], culminating in the release of profibrogenic mediators such as TGF-β as well as pro-inflammatory cytokines such as TNF-α and IL-6 by Kupffer cells [59]. As previously shown, the proinflammatory mediators release, through the activation of the SOCS pathway, could worsen the hepatic IR by inhibiting IRS1 and IRS229. In detail, the interaction between IL-6 and its receptor (IL-6R) located on the hepatocyte membrane leads to the activation of the JAK/STAT3 signaling pathway. Janus-activated kinases (JAKs) determine the phosphorylation of the signal transducer and activator of transcription 3 (STAT3), which, once translocated to the nucleus, induces the expression of the SOCS. Similarly, the interaction between TNF-α and its receptor (TNFR) results in the activation of SOCS. Moreover, differently from IL-6, TNF-α activates SOCS through the JNK-IKKβ-NF-κB pathway. The SOCS-related mechanisms involved in the exacerbation of IR are responsible for the inhibition of I-Rec-mediated IRS1/2 activation. Specifically, SOCS inhibited the tyrosine kinase activity of the insulin receptor, competed with IRS1/2 binding sites on phosphorylated tyrosines, or induced the increased expression of SREBP-1c, which, through a feedback mechanism, suppressed the synthesis of IRS1/2 (Figure 1) [60].

In the NAFLD context, this cascade of events appeared recurrent, as shown in animal and human models. Recently, Carpino et al. revealed an increased liver localization of *Escherichia coli* LPS both in the experimental murine model of NAFLD (high-fat diet (HFD) and methionine–choline-deficient (MCD) mice), as well as in biopsy samples of NAFLD patients, as compared to controls. These highlighted a close relationship between *Escherichia coli* LPS hepatocyte-incremented tropism and the severity of liver damage due to the TLR4 pathway Kupffer cells activation [61,62].

Accordingly, several other studies using TLR4 knock-out mice showed a decrease in lipid accumulation following an MCD or high-fructose diet, as compared to TLR4 wild-type mice, providing further evidence on the role of LPS in the disease progression [63,64].

However, in addition to the above-presented indirect mechanisms involving the LPS-related release of pro-inflammatory cytokines, many other LPS/TLR4–MyD88 pathways interfere with the insulin-signaling cascade in several ways, worsening hepatic IR.

A crucial point was represented by the enhancement of TLR4-mediated de novo ceramide synthesis, which, in turn, led to protein kinase C-θ (PKCθ) and protein phosphatase 2A (PP2A) activation. PKCθ interfered with the activation of IRS1 and IRS2, mediating their phosphorylation at a serine residue, which consequently resulted in their ubiquitination and further proteasomal degradation [58,65]. The mechanism in which PP2A was involved conversely occurred further downstream in the insulin-signaling pathway and was represented by the inhibition of Akt phosphorylation (Figure 1) [60,61]. Furthermore, TLR4 activation induced by LPS led to the increased expression of inducible nitric oxide synthase (iNOS) [66]. The overexpression of iNOS determined the S-nitrosation/S-nitrosylation of protein in insulin-sensitive tissues, which appeared to have a central role in inducing ER stress and IR [67,68]. Zanotto et al. showed, in a model of iNOS knockout, HFD-fed mice, the crucial role of iNOS in regulating muscle insulin sensitivity. In the liver and adipose tissue, the onset of IR caused by HFD was only partially correlated with iNOS activation. Indeed, even in the presence of a genetic or pharmacological blockade of iNOS, ER stress remained strongly associated with altered insulin signaling. Remarkably, pharmacological ER-inhibition through sodium phenylbutyrate (PBA), as well as iNOS inhibition, improved insulin signaling with the complete recovery of glucose tolerance in these tissues [69].

Altogether, these findings suggest a possible combined therapeutic action targeting iNOS-dependent pathways as well as iNOS-independent pathways, in order to improve insulin sensitivity in the liver.

### 2.3. Genetic and Epigenetic Landscape in NAFLD and IR: Molecular Aspects

#### 2.3.1. Genetics

Although the progression of NAFLD relies on metabolic aspects, genetics may play a pivotal role in the establishment of the disease from a pathogenetic-to-clinical point of view [70]. Among the different genetic alterations, single nucleotide polymorphisms (SNPs) of key genes involved in the regulation of lipid and retinol hepatocytes metabolism represented the strongest genetic predictors of NAFLD [71]. A number of genes have been described in this setting, and most were represented by patatin-like phospholipase domain-containing 3 (PNPLA3), membrane-bound o-acyltransferase domain-containing 7 (MBOAT7), and transmembrane 6 superfamily member 2 (TM6SF2) (Table 1). PNPLA3 gene codified for a 481-aminoacid membrane lipase and located on the lipid droplet (LD) surface in hepatocytes, adipocytes, and in hepatic stellate cells (HSCs), to express triglyceride (TG) hydrolase activity [72]. Its expression was regulated by sterol regulatory element-binding protein 1 (SREBP1c)/liver X receptor (LXR) and by carbohydrate response element-binding protein (ChREBP), and activated by post-prandial or pathological hyperinsulinemia [73].

In 2008, a large multi-ethnic population in a genome-wide association study (GWAS) conducted in North America shed some light on the most common gene variant involved in the susceptibility to NAFLD in Hispanics [77]. The rs738409 C > G variant in the PNPLA3 gene, encoding the aminoacidic substitution isoleucine to methionine at the position 148 (p.I148M), was considered the most tightly associated variant with hepatic fat accumulation [74]. It has been recently established that the harmful effects of the IR-related I148M variant hyperexpression were due to its interaction with other lipases, such as adipose TG lipase (ATGL)/patatin-like phospholipase domain-containing 2 (PNPLA2), resulting in the impairment of the biological activity by directly interacting with its cofactor, the comparative gene identification-58 (CGI-58) [78]. In addition, the I148M variant abrogates ubiquitylation and proteasomal degradation, resulting in its accumulation on the LD surface and thus impairing TG mobilization and leading to the increase in hepatocytes fat accumulation [79,80].

Interestingly, this polymorphism was not associated with an increased risk for IR. Insulin resistance, evaluated by HOMA-IR, EHC, fasting or post-glucose insulin and glucose concentrations, did not differ between carriers and non-carriers of the gene variant, while the I148M variant had liver fat 73% more frequently than non-carriers [81].

MBOAT7 gene, also known as lysophosphatidyl-inositol acyltransferase 1 (LPIAT1), codified for an enzyme member of the Lands cycle of phospholipid acyl-chain remodeling of the membranes, through sequential deacylation and reacylation reactions [82]. It was highly expressed in human hepatocytes, sinusoidal endothelial cells, immune cells, and HSCs, but less so in cholangiocytes, being particularly localized on the ER and the mitochondria in which fat biosynthesis occurs [83].

In 2015, the common rs641738 C > T variant was identified as a novel mediator of the susceptibility to develop hepatic damage by the first GWAS on the inherited determinants of alcoholic cirrhosis in heavy drinkers [75,84]. The data were also validated by Mancina and Dongiovanni, who showed the association of the rs641738 variant with steatosis severity and with the entire spectrum of liver damage related to NAFLD, including HCC [85,86].

Overall, these results developed the concept that MBOAT7 could be a regulator, not only of hepatic fat accumulation, but also of whole body adiposity. However, Sookoian et al. demonstrated that MBOAT7 was downregulated in NAFLD even independently of the rs641738 polymorphism [87].

Additionally, Dongiovanni et al. demonstrated that MBOAT7 was downregulated both in patients and in rodents in the presence of severe obesity and hyperinsulinemia, independently of genetic backgrounds [86]. In addition, according to Helsley et al., who confirmed MBOAT7 suppression during IR, and Umano et al., who correlated a lower degree of whole-body insulin sensitivity in obese children and MBOAT7, it was possible to highlight a possible link between the expression of MBOAT7 and IR, which will require further study to be definitively established [88,89].

A GWAS conducted in 2014 and established the missense rs58542926 C > T variant in the TM6SF2 gene, which encodes the lysine-to-glutamate substitution at residue 167 (p.E167K) as a determinant of hepatic TG content, serum aminotransferases, and lower serum lipoprotein [90]. TM6SF2 gene codified for a regulator of cholesterol biosynthesis, which acted in hepatic VLDL lipidation and assembly in the ER cisternae and in ER-Golgi compartments [91].

In a large cross-sectional study in a European cohort consisting of 1201 patients, Dongiovanni et al. recently demonstrated a positive association between the TM6SF2 E167K variant and hepatic steatosis, steatohepatitis, and fibrosis, corroborated by a meta-analysis of four studies and 4325 patients, which had also confirmed higher risk in carriers [76].

Interestingly, Musso et al., demonstrated how the TM6SF2 T-allele, encoding the E167K amino acidic substitution, was associated with higher hepatic and adipose insulin resistance, impaired pancreatic β-cell function, incretin effect, higher muscle insulin sensitivity, and whole-body fat oxidation rates [92]. They showed that the TM6SF2 gene variant affected insulin sensitivity and β-cells function by an impaired β-cells incretin secretion. Therefore, a maladaptive response to a chronic daily repetitive metabolic challenge, such as fat ingestion, could link the TM6SF2 C > T variant to liver injury [93,94].

Together with the meaningful role of genetics in the establishment of a close relationship between IR and the worsening of NAFLD, epigenetics has provided an important contribution to the pathology.

#### 2.3.2. Epigenetics

The liver epigenome is largely disrupted across IR-related disease states. Indeed, a EWAS highlighted how the disruption of hepatic insulin signaling by DNA methylation was a leading process involved in NAFLD, with many genes showing a differential methylation and associated changes in gene expression [95] (Table 2).

Furthermore, advanced stages of NAFLD have been associated with a global hypomethylation and concomitant over-activation of a pro-fibrogenic gene program, as exemplified at fibroblast growth factor receptor 2 (FGFR2) [96]. The hypomethylation of FGFR2 was found on 23 CpG sites, promoting its overexpression and the establishment of inflammatory pro-fibrotic niche96. Additionally, a TF-binding motif analysis of NAFLD differentially methylated sites revealed strong enrichments for binding motifs of hepatic regulators of glucose and lipid metabolism such as PGC1a, SREBF2, FOXA1, and FOXA2, and further supporting the concept that appropriate DNA methylation would be necessary for overall hepatic metabolic homeostasis [97].

Several studies have shown associations between the levels and the activity of epigenetic modifiers and IR-related states, suggesting the pervasive disruption of the epigenome [95]. Particularly, Baur J.A. et al. highlighted the role of sirtuin 1 (SIRT1) as a key metabolic sensor in the liver [98]. SIRT1 is a protein involved in the deacetylation of histones and other factors, and it is induced in fasting, leading to the deacetylation of the transcriptional co-activator PGC-1α, the primary regulator of liver gluconeogenesis, and inducing increased gene expression for gluconeogenesis and a concomitant raise in glucose production [99]. A number of studies have reported that, in the liver, SIRT1 had a decreased expression in insulin-resistant cell lines and tissues from HFD-fed rodents, and its loss or inhibition led to IR [100]. Accordingly, Wang R-H. et al., suggested that liver-specific SIRT1-knockout led to the disruption of mTorc2/Akt signaling downstream of the insulin receptor, leading yet again to IR [101].

Non-coding RNAs such as miRNAs (micro-RNA) and lncRNAs (long-non-coding RNA) have been recognized as promising novel biomarkers and therapeutic targets, especially due to their regulatory functions. The dysregulation of miRNAs and lncRNAs activity has been detected in the livers of insulin-resistant patients. Among them, the most descripted one is miR-122, which has been recognized as the dominant miRNA in the liver, and it accounts for 70% of the hepatic miRNA content in mice and 52% of the human hepatic miRNA [102]. It plays a pivotal role in regulating hepatic gene expression, affecting various aspects of cellular activity such as responses to oxidative stress, inflammation, viral infections, but mostly, its activity has been linked to the regulation of lipid metabolism [103].

Long et al. showed that the upregulation of miR-122 was linked with the over-expression of genes involved in de novo lipogenesis, such as fatty acid synthase (FASN) and SREBP1c in HepG2 and Huh-7 cells, treated with free fatty acids (FFA) for 24 h. In addition, the inhibition of miR-122, conducted with an miR-122 inhibitor, resulted in a significant reduction in the expression of the aforementioned genes, consequently leading to a decreased de novo lipogenesis. Dong et al. noted that a dysregulation of this miRNA was implicated in hepatic insulin resistance [104]. Hence, the overexpression of miR-122 resulted in a decreased expression of its target, such as insulin-like growth factor (IGF-1R), which is a part of the IGF-1R/PI3K/Akt signaling pathway, thus promoting IR. Alongside the data collected in vitro, miR-122′s circulating levels were significantly higher among subjects with insulin resistance, as compared to healthy patients. Moreover, patients suffering from MS were characterized by a 160% higher levels of circulating miR-122. These findings indicated that miR-122 was a very promising marker of hepatic insulin resistance, and its negative effect on the insulin pathway could be overcome by miR-122 inhibition.

Recently, another miRNA, miR-499105, has been associated with hepatic insulin resistance and thus with NAFLD by Hanyun et al. The experiments were conducted on specific germ-free (SPF) male C57BL/6 mice fed a high-fat diet and injected with a miR-499 inhibitor. The treatment significantly reduced liver fat accumulation in the treated mice, as compared to the non-treated ones [105].

Additionally, Wang et al. pointed out the meaningful role of miR-499 in hepatic glucose metabolism by showing a significant decrease in miR-499 levels in the livers of db/db and HFD-fed mice [106]. The downregulation of miR-499 contributed to an impairment in Akt/GSK activation. The association between miR-499, the insulin-signaling cascade, and glycogen synthesis was due to the suppression of PTEN biological activity.

In addition to the implications of microRNAs in the management of hepatic insulin resistance, lncRNAs have been reported to play a critical role in this context.

Among them, recent evidence has suggested MALAT1′s effect on glucose and lipid metabolism in several metabolic dysfunctions, such as NAFLD [107]. MALAT1 is a long non-coding RNA well-conserved among different mammal species. According to Yan et al., who investigated the expression of MALAT1 in two models of diabetes including the liver of ob/ob mice and in hepatocytes exposed to palmitate, the expression of MALAT1 was upregulated [108]. Additionally, Chen et al. showed that MALAT1 was involved in oxidative stress-mediated insulin resistance via the upregulation of the c-Jun N-terminal kinase (JNK), a stress-sensitive kinase, that upon activation, could suppress insulin signaling by inhibiting the phosphorylation of IRS and Akt, two major regulators of insulin-signaling cascades [109].

Finally, a remarkable role in hepatic insulin resistance was indicated for long non-coding H19. Nilsson et al. observed the elevation of H19 hepatic levels in adults with type 2 diabetes [110].

The mechanism by which H19 was involved in hepatic glucose homeostasis was related to a knock-down in HepG2 cells followed by an RNA-seq analysis and qPCR, in which the expression levels of HNF4A as well as PCK1 and G6PC, two major gluconeogenic genes, appeared to decrease [110].


Additionally, a recent study conducted by Liu et al. emphasized the correlation between the upregulation of H19 and the over-expression of SREBP-1c, ACC1, SCD1, FASN, and PPARγ in a model of NAFLD, highlighting the crucial role of H19 in the development of hepatic insulin resistance by affecting not only genes involved in gluconeogenesis but also in hepatic lipogenesis [111]. jcm-11-03649-t002_Table 2Table 2Epigenetics mechanisms involved in NAFLD pathogenesis and IR.EpigeneticsTargetBiological SignificanceReferenceDNA hypomethylation↑ FGFR2Pro-fibrogenic[96]↑ PGC1a, ↑ SREBF2,↑ FOXA1, ↑ FOXA2Hepatic regulators of glucose and lipid metabolism[97]↓ SIRT1↑ PGC-1αPrimary regulator of liver gluconeogenesis[99]↓ mTorc2/Akt signalingInsulin resistance[101]↑ miR-122↑ FASN*De novo* lipogenesis[104]
↑ SREBP1c


↓ IGF-1R

↓ miR-499↓ Akt/GSK activationLiver fat accumulation[106]
↓ PTENHepatic glucose metabolism
↑ LncRNA MALAT1↑ JNKOxidative stress-mediated insulin resistance[107]

Effect on glucose and lipid metabolism[109]↑ LncRNA H19↑ SREBP-1c, ↑ ACC1, ↑ SCD1, ↑ FASN, ↑ PPARγDevelopment of hepatic insulin resistance[111]Up arrows (**↑**) and down-arrows (**↓**) indicate upregulation and downregulation, respectively, of epigenetic mechanisms related to their specific targets.

The recent escalation of metabolic disorders in Western countries has drawn attention to the correlation between these diseases and changes in dietary habits. The Western diet has been modified, replacing fruits, vegetables, proteins, and omega-3 fatty acids with saturated and trans fats, omega-6 fatty acids, carbohydrates, and high-energy nutrients [112]. It has been shown how a change in lifestyle may improve NAFLD and its comorbidities. From this perspective, nutrigenomics appears to be crucial in this setting, particularly considering the potential benefit that could be derived from dietary therapeutic application and tailored clinical management [113]. Nutrigenomics examines the interaction of dietary exposure with the genome, and it is focused on the observation of gene-diet effects that may affect health; thus, it represents a fascinating discipline [114]. In this regard, fructose-enriched foods are considered one of the major categories of food directly involved in NAFLD onset and evolution due to its role as a substrate for DNL, which increased the acetil-CoA cellular level and acted as an enhancer for the expression of several enzymes involved in DNL, such as SREBP-1c and ChREBP [115]. In addition, it also has been hypothesized that the intermediate products, such as diacyglycerols generated when fructose is converted to triglycerides, could trigger insulin resistance and subsequent fat accumulation in the liver [116]. Notwithstanding, in pre-clinical models, an augmented consumption of n-3 PUFAs in the daily diet has been correlated to a suppression of diet-induced steatosis and an improvement in IR as a consequence of a decreased activity of (SREBP)-1 transcription factor and fat uptake genes, as well as by increasing the expression of genes involved in FAs oxidation [117].

Despite this evidence, which shed light on a new field closely linked to the management of metabolic diseases including NAFLD, there are still many biological mechanisms to be addressed such as revealing persistent genome-nutrient interactions that may affect both clinical and health nutrition practices.

### 2.4. Role of Other Hormones in Fueling IR: A Matter of Sex?

The liver has a pivotal role in the maintenance of glucose and lipid homeostasis as well as in the regulation of energy metabolism. Glucose homeostasis is preserved through the regulation of glycogen breakdown/synthesis, glycolysis and gluconeogenesis (GNG), respectively [118]. The liver behavior differs between the sexes in terms of the management of glucose equilibrium. Indeed, females are more susceptible to insulin action in the liver, as compared to males, by reducing the hepatic glucose production at a low insulin concentration [119]. The main trait for the discrepancy between males and females’ liver function was correlated to its metabolic adaptation to fasting and refeeding conditions and to the regulation of glucose homeostasis, a feature that could partly explain the sex-specific susceptibility to type-2 diabetes (T2D) [120,121]. Several studies pointed out the meaningful role of estrogens and their receptors in the hepatic sexual dimorphism as well as in the sex differentiation of the liver [122,123].

Estrogens act by binding their most expressed receptor in the liver, ERα, which after dimerization, activates the estrogen response element (ERE) in the promoters of target genes [124,125]. In both males and females, estrogen signaling improved glucose tolerance and insulin sensitivity and may be considered the key factor between the two sexes in the sex-specific regulation of glucose homeostasis and, in turn, in many metabolic disorders such as T2D or NAFLD [120,126].

Estrogens suppressed gluconeogenesis and improved insulin sensitivity acting via FOXO1 (forkhead transcription factor 1), a transcription factor with a pivotal role in HGP [127].

Although the specific role of estrogen in the modulation of glucose equilibrium remains unclear, sex differences may rely on estrogens signaling. As compared to males, pre-menopausal females showed higher glucose tolerance and greater insulin sensitivity, as reported by Mauvais-Jarvis in 2018 [120]. Additionally, estrogen deficiency predisposed post-menopausal females to dysglycemia and to impaired hepatic insulin clearance, which could be improved by estrogen administration, thus underlining the important function of estrogen in this context. The hormone was also involved in the regulation of hepatic lipid metabolism, as demonstrated by several pre-clinical and clinical studies [122,128]. In the female liver, under physiological conditions, estrogen signaling inhibited the expression of genes involved in de novo lipogenesis (DNL), such as SREBP-1C, FASN, and SCD1, and promoted FAO [129]. At the molecular level, estrogens regulated hepatic lipid metabolism, primarily acting through ERα, which is the predominant ER subtype in both male and female hepatocytes [123]. Indeed, male and female ERαKnock-out mice developed hepatic steatosis as a consequence of the increased expression of genes involved in DNL (e.g., SREBP-1c, Scd1) and the decreased expression of genes involved in lipid transport, as demonstrated by Bryzgalova in 2006 [130]. Furthermore, the lack of estrogen in females, caused by menopause or by polycystic ovary syndrome (PCOS), may trigger an altered hepatic lipid metabolism with a higher risk of developing NAFLD [131]. Although not fully investigated, the role of estrogens may be crucial for males, as well, as the lack of ERα has led to increased hepatic lipid droplets and TG content, as a result of enhanced DNL [132]. By contrast, androgens, produced by males and females, have sex-dependent differences in their behaviors [133]. It has been shown that the hepatic expression of androgen receptors (AR) was several-fold higher in males than in females, increased with puberty, and gradually declined with age. AR was expressed by hepatocytes but not by other hepatic cell types such as KCs and liver sinusoidal cells [134].

Androgens play a relevant role in the glucose homeostasis as described by several studies, which emphasized that altered androgen levels increased hepatic glucose output, induced hyperglycemia, and led to a high risk of developing T2D in both males and females [135].

Under physiological conditions in males, testosterone increased insulin receptors through AR, thus decreasing glucose uptake by inhibiting the transcription and translation of GLUT-2, as well as inducing the synthesis of glycogen. Testosterone was also responsible for the repression of gluconeogenesis through the interaction with FOXO1, which led to an improvement in glucose metabolism [136]. Conversely, in females, pathological conditions that determine an excess of androgens, such as polycystic ovary syndrome (PCOS), impaired hepatic glucose metabolism by decreasing insulin-stimulated glucose uptake and glycogen synthesis, thus predisposing PCOS females to insulin resistance [137]. In addition, this hormone was also implicated in lipid metabolism as male androgen-signaling is involved in the regulation of lipid metabolism by inhibiting the expression of crucial genes implicated in DNL such as SREBP-1C and PPARγ and by promoting FAO through the upregulation of PPARα [138]. The sex-specificity of this hormone was identified in this process. In this regard, although beneficial in males, excessive androgen impaired hepatic lipid metabolism and promoted hepatic steatosis in androgen-treated female rodents, as showed by Seidu et al. in 2021 [139], which was similar to the results obtained in PCOS females (Figure 2).

Altogether, these findings suggest the importance of sex-hormones in the incidence of NAFLD. Initially, females appeared to be less susceptible to NAFLD before menopause. The condition was reversed in post-menopausal females with low levels of estrogen, which normally had a protective effect against the progression of NAFLD [140]. Moreover, an augmented risk in the onset of NAFLD was found in young females suffering from reproductive dysfunction characterized by altered estrogen levels (such as PCOS, Turner Syndrome), as compared to young fertile females [141]. Additionally, Kamada et al. highlighted how in ovariectomized (OVX) female mice fed with a high-fat diet (HFD), prolonged estrogen deficiency boosted hepatic inflammation and worsened liver injury, as well as in post-menopausal females with NAFLD [142]. After menopause, the lack of estrogens favored the redistribution of fat towards visceral deposits and relieved the inhibition of adipose lipolysis, boosting the FFA flux to the liver and increasing the risk of developing NAFLD via IR impairment [143]. A study conducted in 2017 on obese males pointed out the correlation between low serum testosterone levels and IR and erectile dysfunction in patients with NAFLD, as compared to the males with normal BMIs [144]. Moreover, a reasonable explanation for the role of IR in ER was found by Kelly et al., who showed that IR led to enhanced vascular expression of endothelin-β receptors that then contributed to augmented ROS production, endothelial dysfunction, and increased vasoconstriction in erectile tissue from insulin-resistant obese rats [138]. Therefore, it may be logical to state that the treatment of IR may carry a dual benefit of improving erectile function and decreasing the grade of hepatic steatosis. In addition, other hormones have been implicated in the regulation of glucose and lipid metabolism, hence sustaining basal energy expenditure, including thyroid hormones [145]. Thyroid hormones (TH) include tri-iodothyronine (T3) and 3,3′,5,5′-tetraiodo-L-thyronine (T4), which are required for metabolism balance. T3 controls the expression of genes involved in hepatic lipogenesis and the genes involved in the oxidation of free fatty acids through the thyroid hormone receptor-β (THR-β), which represents the main isoform expressed in the liver [146]. It has been reported that in THR-β-knockout animals, there was an increase in liver mass and hepatic lipid accumulation through increased lipogenic genes and decreased fatty acid β-oxidation [147]. In addition, an augmented hepatic lipid deposition induced the downregulation of several metabolism-related genes, which are dependent of T3 actions [148]. Furthermore, T3 stimulates enzymes that catalyze several important steps in hepatic fatty acid synthesis, such as acetyl-CoA carboxylase (which catalyzes the carboxylation of acetyl-CoA to malonyl-CoA, the first step in hepatic fatty acid synthesis) and fatty acid synthetase [149]. In addition, this hormone is also responsible for the hepatic mitochondrial turnover by increasing mitochondrial biogenesis and mitophagy [148]. Hypothyroidism has been associated with hyperlipidemia through modifications in lipid synthesis, absorption, circulation, and metabolism, as well as with impaired glucose and insulin metabolism, two major risks factors for NAFLD onset [150,151]. Considering the intense relationship between hormones and the development of NAFLD, further research is needed to provide a better understanding of this complex pathophysiological picture.

## 3. Insulin Resistance as the Cornerstone for the Clinical Management of NAFLD: An Overview on Diagnosis, Prognosis, and Potential Treatment Implications

As discussed, IR constitutes the common denominator of the systemic molecular mechanisms sustaining NAFLD genesis and progression, and the common thread unifying its related extrahepatic metabolic comorbidities. From a clinical point of view, according to the European and Asiatic guidelines on this topic, the assessment of IR represents a fundamental pillar in the diagnosis and prognosis stratification of NAFLD patients [2,152].

Consistent also with the recent consensus-proposed redenomination of MAFLD 6, the diagnosis has been founded on the demonstration of an excessive hepatic fat accumulation associated with a status of IR [2,152]. A variety of dynamic techniques commonly employed in diabetes research and clinical management exist to assess insulin sensitivity in routine practice: the oral glucose-tolerance test (OGTT), the insulin-suppression test (IST), and the hyperinsulinemic-euglycemic clamp (HEC). However, despite their high reliability and reproducibility, they are time-consuming and expensive tests, which impact their usage by clinicians [153]. For this purpose, several surrogates have been studied to define the IR grade. Among these, the homeostasis model assessment of insulin resistance (HOMA-IR) has been proposed as the classic surrogate for IR and an acceptable alternative to dynamic methods [2]. Nevertheless, recent evidence suggested limitations in this model: its validity depends on the ability of insulin secretion to adapt to IR, questioning its suitability in overt diabetes. In addition, the assays for insulin measurements vary widely, and there was no agreement on a threshold defining IR when using HOMA-IR [2]. Therefore, with particular reference to the IR-related NAFLD, modern research applications have focused on the identification of new surrogate markers for IR. A recent study in a cohort of Asian patients evaluated the efficiency of the triglyceride glucose (TyG) index, providing a median cut-off of 8.5 and revealing its superiority, as compared to the HOMA-IR, in predicting NAFLD [154]. Furthermore, the inclusion of serum level triglycerides in this modern model confirmed the nature of the disease as a systemic metabolic dysfunction. From a prognostic point of view, considering the IR capability to fuel the disease progression to more advanced stages of NASH and fibrosis, the routine assessment of insulin sensitivity in NAFLD patients could be essential for the stratification of risk [155]. Not surprisingly, some of the non-invasive scores for the gradation of fibrosis, such as the NAFLD fibrosis score (NFS), included the evaluation of the presence/absence of IR and/or diabetes in their algorithms [156].

Furthermore, in the light of implications for NAFLD onset and worsening, lifestyle changes (e.g., adequate physical activity and nutritional habits typical of a Mediterranean diet) continue to represent a first-line approach proposed to these patients with tangible positive effects on insulin sensitivity [157]. Regarding further treatment in non-compliant patients and in more advanced stages of the disease (NASH/fibrosis), the previously described IR-related molecular mechanisms represent a potential target for the use of agents already available in the management of other IR-related metabolic manifestations, particularly type-2 diabetes mellitus (TD2M), as well as an open research challenge for the identification of novel drugs (Figure 3).

Metformin, an insulin-sensitizing agent primarily used in the treatment of TD2M patients, was able to reduce hepatic gluconeogenesis including the enhanced activation of the AMP-activated protein kinase and/or the interference with cell (cytosolic and mitochondrial) redox state, though its mechanisms of action remain unclear [158]. However, two meta-analyses revealed metformin’s failure to improve biochemical outcomes or NASH histology [159,160]. Therefore, metformin-based monotherapy is currently not recommended for the treatment of NAFLD.

At the hepatic and systemic level, the activation of the peroxisome proliferator-activated receptor PPAR-α/δ has led to several consequences, including regulating peroxisomal ß-oxidation of FFA in the improvement of insulin sensitivity and antiphlogistic effects [161]. In the GOLDEN-505 trial with NASH patients (n = 247) treated with the PPAR-α/δ agonist Elafibranor (80 mg or 120 mg), as compared to placebo, for 52 weeks, 162 could not reach the histological resolution of NASH, but a post hoc analyses in a sub-group with more pronounced inflammation revealed the treatment’s association (120 mg) with an improvement in HOMA-IR, plasma triglyceride, and FFAs levels [162].

Furthermore, the farnesoid X receptor (FXR) activation reduced liver lipid levels by inducing the repression of genes involved in lipogenesis (SREBP-1) and gluconeogenesis (ChREBP) [163]. In the phase IIb multi-center, randomized, double-blind, placebo-controlled trial (FLINT) comparing the administration of 25 mg of the FXR agonist obeticholic acid (OCA) versus placebo, OCA was associated with a significant improvement in the NAFLD activity score and a significant improvement in the fibrosis stage in the treated group (35% vs. 19%, *p* = 0.004) [164]. A phase III trial (REGENERATE) is currently evaluating the potential administration of a lower dose of OCA in NASH patients for equal efficacy and greater tolerability (NCT02548351) [165].

In terms of the kidney, the sodium-glucose transporter 2 (SGLT2) is responsible for the reabsorption of a large portion (approximately 90%) of glucose [166]; based on this rationale, SGLT2 inhibitors (“glifozin”) represented a novel pharmacological frontier in the TD2M, in which these agents have been shown to also prevent cardiovascular events [167]. In consideration of this finding, a large RCT recently evaluated the use of SGLT2 inhibitors in patients with T2DM and NAFLD, reporting the empagliflozin association with improvements in hepatic steatosis (in terms of fat accumulation evaluated using magnetic resonance imaging) without evaluating histological features [168]. Finally, considering the gut microbiota capability to influence several NAFLD pathogenetic mechanisms, the modulation of the gut microbiome could be a promising strategy for the management of IR-NAFLD when using probiotics. A meta-analysis comparing four RCTs found that probiotics reduced liver aminotransferases and HOMA-IR scores, among other parameters assessed [169]. However, the possibility of having valid probiotics in clinical practice for patients with IR-NAFLD is strictly dependent on the knowledge and characterization of an individual patient’s intestinal microbial composition to determine a specific disease phenotype. In an era of precision medicine and tailored therapies, the influence of the individual genetic backgrounds on the response to specific treatments must be considered [170].

## 4. Conclusions

Considering the metabolic revolution of chronic hepatopathies in this dynamic field, an analysis of the molecular aspects behind NAFLD pathogenesis could be crucial to define the foundation for future management and treatment development. A hybrid strategy that considers new findings and old knowledge regarding IR represents the cornerstone of a tailored approach for patients and is much needed in the current frontier of hepatology. In this field, the identification and the clarification of the pathogenetic mechanisms underlying IR and fueling the genesis and progression of NAFLD as potential new targets for innovative drugs do not represent a challenge aimed only at expanding knowledge and literature on this argument, but a concrete goal for international research to improve the prognosis and quality of life for these patients.

## Figures and Tables

**Figure 1 jcm-11-03649-f001:**
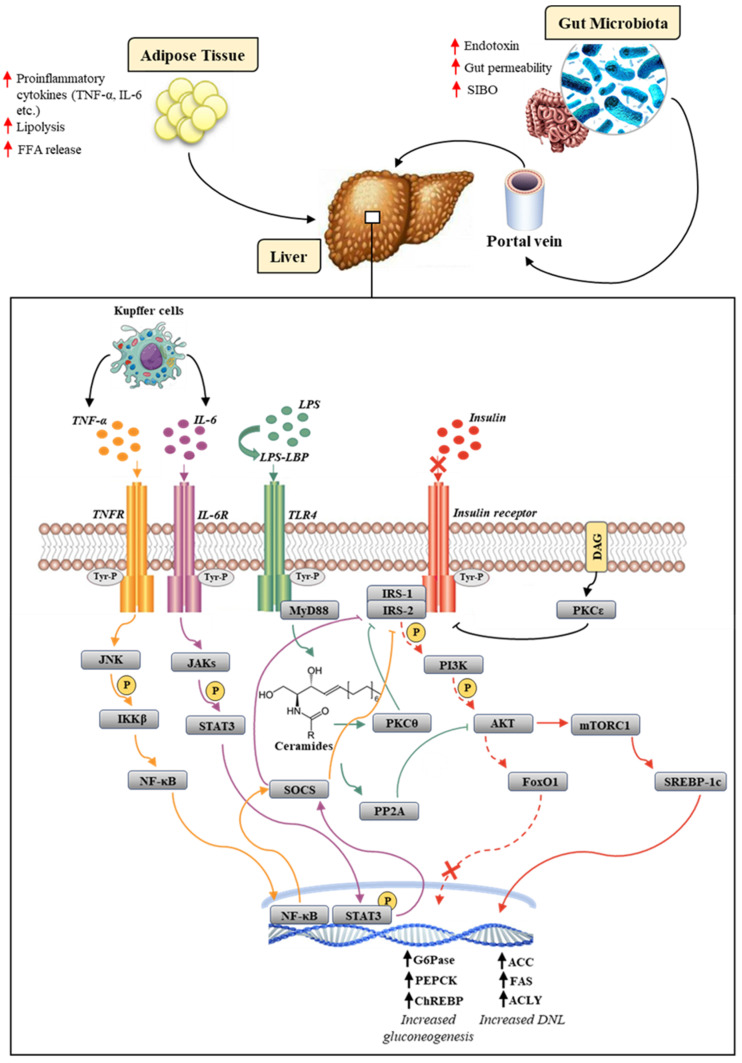
Overview of molecular mechanisms underlying worsening NAFLD pathogenesis and IR. The insulin-signaling cascade is abolished in the Akt/FoxO1 pathway (dashed red arrow), but not in the mTORC1/SREBP-1c pathway (solid red arrow). The LPS-TLR4 interaction culminates in the activation of PKCθ, which mediates the proteasomal degradation of IRS1/2 and PP2A, which inhibits Akt further downstream in the insulin cascade (green arrow). The release of cytokines, such as IL-6 and TNF-α, from Kupffer cells results in the activation, following the interaction with their own receptors, of the JAKs/STAT3 (purple arrow) and JNK/NF-κB (yellow arrow) pathways, respectively. The latter both culminate in the activation of SOCS, which is responsible for inhibiting IRS1/2. The diacylglycerol (DAG) activates PKCε, which inhibits the tyrosine kinase activity of the insulin receptor (black arrow). Up arrows (↑) stand for enhanced expression (in the case of a gene), production (in the case of mediators), and occurrence (in the case of the relative pathological event).

**Figure 2 jcm-11-03649-f002:**
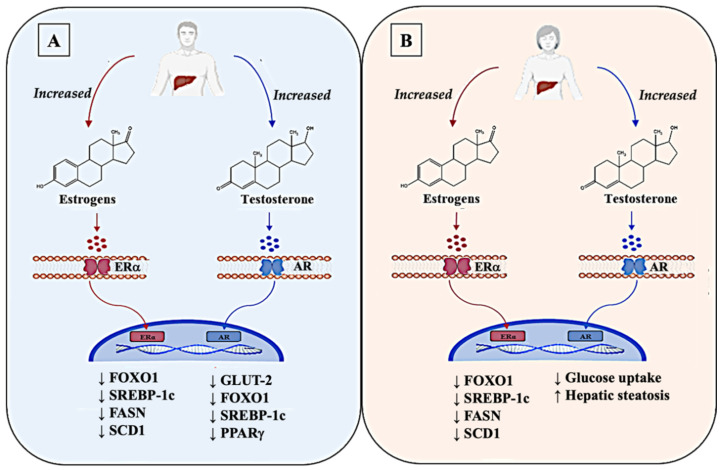
Dimorphism of the sex hormones estrogens and testosterone in the liver of the two sexes. Estrogens display a similar role in both sexes, interacting with their receptors (ERα) in plasma membrane of liver cells, which, once activated, migrate in the nucleus and bind to estrogen-responsive element (ERE) on their target genes. Estrogens suppress hepatic glucose production (HGP) by inhibiting the expression of FOXO1. Estrogens also impair the hepatic de novo lipogenesis (DNL) by reducing the expression of genes such as SREBP-1C, FASN, and SCD1, and promote fatty-acid oxidation (FAO). Overall, in physiologic conditions, estrogens play a significant role in the modulation of glucose equilibrium in the liver of both sexes (**A**,**B**). In contrast, androgens, especially testosterone, show different functions in the liver of the two sexes through the interaction with their membrane receptors AR. In male livers, an increase in testosterone levels has beneficial effects in reducing glucose uptake, by inhibiting the transcription of GLUT-2. It has an important action also in the reduction in DNL in male livers, acting in the suppression of the transcription of SREBP-1C and PPARγ and in the inhibition of HGP through FOXO1 (**A**). Differently, in female livers, altered levels of testosterone promote an increase in hepatic steatosis by impairing hepatic glucose metabolism, thus predisposing females to insulin resistance (**B**). Up arrows (↑) and down arrows (↓) indicate, respectively, enhanced and reduced expression (in the case of a gene), production (in the case of mediators), and occurrence (in the case of the relative pathological event).

**Figure 3 jcm-11-03649-f003:**
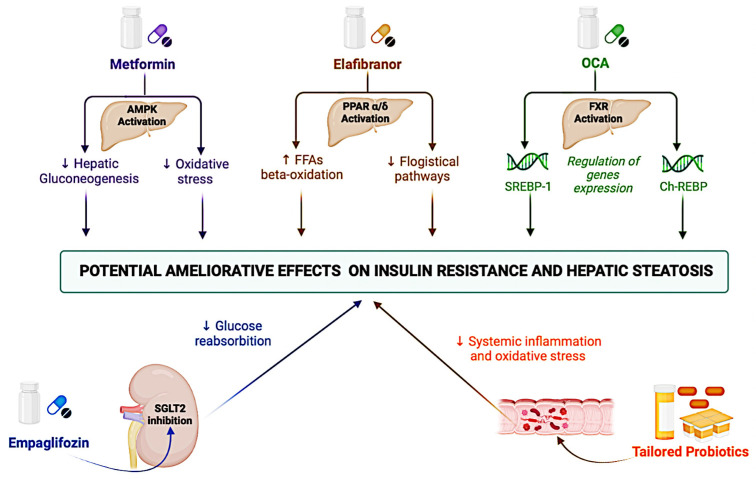
Principal mechanisms of action of therapies currently undergoing clinical studies for ameliorating IR-related hepatic steatosis. AMPK: AMP-activated protein kinase; ChREBP: Carbohydrate Response Element Binding Protein; FXR: Farnesoid X receptor; FFAs: Free Fatty Acids; IR: Insulin resistance; OCA: obeticholic acid; PPAR: Peroxisome proliferator-activated receptor; SGLT2: Sodium-glucose transporter 2; and SREBP-1: Sterol regulatory element-binding protein 1.

**Table 1 jcm-11-03649-t001:** Genetics mechanisms involved in NAFLD pathogenesis and IR.

Gene	Predisposing Variant	Biological Significance	Reference
PNPLA3	rs738409 C > G	Hepatic fat accumulation	[74]
MBOAT7	rs641738 C > T	Susceptibility for hepatic damage	[75]
TM6SF2	rs58542926 C > T	Hepatic steatosis, steatohepatitis, and fibrosis	[76]

## Data Availability

This study did not report any data.

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
