# Peer review of "The Role of Insulin Resistance in Fueling NAFLD Pathogenesis: From Molecular Mechanisms to Clinical Implications"

_jcm, 2022, doi:10.3390/jcm11133649_

Round 1

Reviewer 1 Report

Palma et al. proposed a very interesting and a very complete paper, reviewing the different pathogenesis mechanisms associated with NAFLD occurence. They also made a link with some major trials related to theses pathways (This is a straigth selection, it is hard to enlarge, I suppose.).

I noted, nevertheless, that a global scheme would be helpful.

Also, some syntax mistakes can happen (lines 131, 169, 217, 232, 234....).

To summarize, I am enthsiastic.

Author Response

We thank the reviewers for appreciating the manuscript entitled “The role of insulin resistance in fueling NAFLD pathogenesis: from molecular mechanisms to clinical implications” written by Palma et. al and submitted to the section “Gastroenterology & Hepatopancreatobiliary Medicine”, as well as for the valuable suggestions and minor revisions proposed.

We have corrected the syntax errors throughout the text.

Regarding the paragraph relating to drugs currently under study in clinical trials, as brilliantly supported by the reviewer 1, the choice to describe the specific ones aimed to report the larger and more advanced trials as well as the ones analyzing the drugs having potential effects on the pathogenetic mechanisms treated in the previous paragraphs. However, as suggested by the reviewer, a global scheme would be useful for the reader. Therefore, we have added a figure (figure 3) showing in an extremely simple and intuitive way the main mechanisms of action of the drugs currently under study for this condition.

Reviewer 2 Report

This article provides a relatively comprehensive review of the role of IR-related molecular mechanisms on the pathogenesis of NAFLD, including the “classic” and more innovative models, and the potential clinical value based on the novel findings. The paper is clearly structured and rich in content, with reference value for researchers and clinicians in this field.

Small editing errors need to be modified, see the yellow marking section in the text.

Author Response

We thank the reviewers for appreciating the manuscript entitled “The role of insulin resistance in fueling NAFLD pathogenesis: from molecular mechanisms to clinical implications” written by Palma et. al and submitted to the section “Gastroenterology & Hepatopancreatobiliary Medicine”, as well as for the valuable suggestions and minor revisions proposed.

We have corrected the syntax errors throughout the text.

Reviewer 3 Report

The author made an overview of molecular mechanisms underlying NAFLD pathogenesis and the important role of insulin resistance. The manuscript is very interesting and well-writen. Here are some minor concerns.

1. The authors should better highlight the purpose of the review and the impact on scientific community 

2. Please explain ↑↓ in the figures and tables.

3. Higher resolution images should be provided.

4. The expression of microorganisms should be in italics.

Author Response

We thank the reviewers for appreciating the manuscript entitled “The role of insulin resistance in fueling NAFLD pathogenesis: from molecular mechanisms to clinical implications” written by Palma et. al and submitted to the section “Gastroenterology & Hepatopancreatobiliary Medicine”, as well as for the valuable suggestions and minor revisions proposed.

We improved the resolution of the figures and clarified the meaning of the up and down arrows for both images and tables. Microorganisms are now written in italics font.

Finally, we have better defined the purpose of our work and the clinical implications that may derive from it.